# The Effect of Viral Clearance Achieved by Direct-Acting Antiviral Agents on Hepatitis C Virus Positive Patients with Type 2 Diabetes Mellitus: A Word of Caution after the Initial Enthusiasm

**DOI:** 10.3390/jcm9020563

**Published:** 2020-02-19

**Authors:** Davide Giuseppe Ribaldone, Marco Sacco, Giorgio Maria Saracco

**Affiliations:** Gastro-hepatology Unit, Department of Medical Sciences, University of Turin, 10126 Turin, Italy; marco.sacco10@gmail.com

**Keywords:** Hepatitis C virus, chronic hepatitis C, insulin resistance, diabetes mellitus, impaired fasting glucose, direct-acting antiviral agents, cirrhosis, antidiabetic therapy

## Abstract

The causal link between chronic hepatitis C and glycometabolic alterations has been confirmed by much biochemical, clinical, and epidemiological research work, but what is still controversial is the long-term clinical impact of sustained virologic response (SVR) achieved by direct-acting antiviral agents (DAAs) on patients with type 2 diabetes mellitus (DM). The aim of this paper is to summarize the biochemical and clinical consequences to DM of DAA-based therapy for hepatitis C virus (HCV) infection. An electronic search of Embase, PubMed, MEDLINE, Ovid, and the Cochrane Database of Systematic Reviews was conducted for publications assessing whether clearance of HCV achieved by interferon (IFN)-free antiviral therapy determines significant changes in glycometabolic control and clinical outcomes of diabetic patients. A beneficial effect of SVR obtained by DAA therapy on DM prevention and the short-term outcome of glycometabolic alterations are acknowledged by most of the studies. Whether this effect is maintained over the long term with a significant clinical impact on diabetic and liver disease is still a matter of debate.

## 1. Introduction

Hepatitis C virus (HCV) infection has been associated with the development of extrahepatic manifestations [1], including type 2 diabetes mellitus (DM). Several reviews and meta-analyses [2,3,4,5] have shown a higher prevalence of DM in patients with chronic hepatitis C (CHC) than in controls. The processes by which HCV induces DM have not been completely understood, but evidence [6,7,8,9,10] suggests that insulin resistance (IR) is the main mechanism, triggered by several pathogenic causes such as proinflammatory cytokines, chemokines, and immune-mediated actions leading to the final impairment of insulin receptors (Figure 1). Interestingly, whole-body insulin sensitivity is affected in patients with CHC [11], but antiviral therapy with direct-acting antiviral agents (DAAs) seems to predominantly improve peripheral insulin sensitivity [12].

According to a recent meta-analysis [13], in patients with CHC treated with interferon (IFN), the achievement of a sustained virologic response (SVR) significantly reduces the incidence of IR during follow-up; this is reflected in decreased frequency of DM in the years following viral eradication.

What is still controversial is the role played by SVR in HCV-positive patients with glucose intolerance (impaired fasting glucose, IFG) or established DM. In fact, the mechanism by which viral eradication would determine an improvement of the biochemical parameters of HCV-positive patients with DM or IFG has not yet been clarified. In addition to the uncertainty about how viral eradication would act on glycemic dysmetabolism, there is a lack of prospective studies with appropriate follow-up on the outcomes of diabetic patients achieving SVR. With the advent of DAAs and consequent high SVR rates, several observational studies [14,15,16,17,18,19,20,21,22,23,24,25,26,27] and one meta-analysis [28] suggested a significant clinical impact of SVR on DM outcomes; however, two of them [18,19], despite reporting significant initial amelioration, did not observe long-term improvement in glycemic control and some other studies [29,30,31,32,33] failed to find any significant glycometabolic changes. Controversies still exist regarding the post-SVR outcome of DM; current data are not sufficient to determine whether the acquired effect observed at SVR (significant decrease in fasting glycemia and glycated hemoglobin, HbA1c) persists over time and if it induces relevant variations regarding antidiabetic therapy. Finally, the influence of DAA-induced SVR on the long-term outcome of DM in diabetic patients remains largely unknown.

In order to address these issues, we performed a systematic review of pertinent studies published up to 30 June 2019.

## 2. Experimental Section

An electronic search of Embase, PubMed, MEDLINE, Ovid, and the Cochrane Database of Systematic Reviews was conducted for publications, with no language and study type restrictions, up to 30 June 2019. The keywords used were: chronic hepatitis C, diabetes mellitus, diabetes, type 2 diabetes mellitus, insulin resistance, cardiovascular diseases, renal insufficiency, retinopathy, microangiopathy, hepatitis, chronic hepatitis C, hepatitis C virus, HCV, meta-analysis, systematic review, review, direct-acting antiviral agents, cirrhosis, hepatocellular carcinoma (HCC). All the articles obtained were evaluated and accepted if they met one or more of the following criteria: (1) data on the incidence of IR and/or DM in the follow-up of patients treated with DAAs; (2) sufficient data to establish the glycometabolic response in HCV-positive diabetic patients after DAA therapy; (3) data on the incidence of DM-related events in the follow-up of diabetic patients treated with DAAs; and (4) data on the incidence of liver-related events in the follow-up of diabetic patients treated with DAAs. We excluded studies that involved patients without IR and/or DM and conference abstracts.

Data were independently extracted by two researchers (D.G.R. and M.S.) and discrepancies were resolved by the third reviewer (G.M.S.). The following information was collected from each study: first author, year of publication, country of origin, study and design type (randomized controlled double blind/open label trial, observational cohort prospective/retrospective study, cross-sectional study, case report), control group (untreated patients or nonresponders), total sample size, gender, age (mean and range), glycemic and HbA1c levels pre/post DAA therapy and during post-SVR follow-up, length of post-SVR follow-up, number and percentage of patients with reduced antidiabetic therapy, and number and type of DM-related and liver-related events in diabetic patients during post-SVR follow-up.

Five issues regarding glycometabolic control in diabetic patients with CHC who were successfully treated with DAAs were identified, and the following pertinent questions were posed:(1)Does SVR achieved by DAAs significantly prevent the onset of IR and DM?(2)Does HCV clearance with DAAs lead to significant improvement of glycometabolic control in patients with DM? If so, is this control maintained over the long term?(3)Does SVR-related glycometabolic improvement induce de-escalation/withdrawal of antidiabetic therapy?(4)Do DAA-induced glycemic changes determine a significant clinical impact on the outcome of DM and its complications?(5)What is the clinical impact of DAA-induced SVR on the incidence of HCC in diabetic patients?

## 3. Results and Discussion

### 3.1. Does SVR Achieved by DAAs Significantly Prevent the Onset of Insulin Resistance and DM?

In the pre-DAA era, several reviews and meta-analyses [13,34,35] showed that SVR impacted the incidence of IR and DM in cured patients. This observation was confirmed by most of the studies conducted on patients who were treated with DAAs (Table 1).

Adinolfi and colleagues [38] prospectively evaluated the changes concerning IR in nondiabetic patients with HCV genotype 1 and advanced hepatic fibrosis treated or untreated with DAAs. Patients achieving SVR showed a significant reduction in blood glucose and insulin levels and a concomitant improvement in insulin sensitivity, which did not occur in untreated patients. Improvement of IR appeared independent of fibrosis severity and body mass index (BMI) value. Gualerzi et al. [39] prospectively analyzed 82 HCV-RNA-positive nondiabetic patients, most with cirrhosis, who were treated with DAAs. Before and immediately after antiviral therapy, all underwent a standard oral glucose tolerance test (OGTT); the OGTT-derived indices were homeostasis model assessment (HOMA)-IR and insulin sensitivity index (ISI). The authors showed baseline glucose metabolism abnormalities in about 55% of patients; significant improvements of HOMA-IR and ISI were found in both prediabetic patients and normal glucose-tolerant patients. Salomone and colleagues [36] studied 32 patients with genotype 1 and cirrhosis treated by DAAs, all with OGTT-documented IFG. After therapy, all patients achieved SVR and significant improvement of the glycemic curve and ISI, despite there being no change of BMI and lifestyle. Similar results were described by a recent Brazilian study [40] that monitored HOMA-IR to verify changes in IR in patients with CHC treated with DAAs. HOMA-IR was significantly improved only in nondiabetic patients with a pretherapy HOMA-IR > 2.5. According to Elhelbawy and colleagues [37], a statistically significant difference in 12-week post-treatment HOMA-IR was found in 465 sustained responders compared to 46 nonresponders (1.9 versus 3.33, *p* = 0.001). Contradictory results were reported by an American [41] and a Canadian [42] study: In the first, 60 HCV-positive patients, genotype 1 (most of them nondiabetic or prediabetic), were treated with sofosbuvir and ribavirin for 24 weeks and followed for an additional 24 weeks post-therapy; SVR was achieved by 70% of patients. According to the authors, HOMA-IR did not significantly change in either sustained responders or nonresponders. However, the results may have been affected by the incomplete adherence to fasting in blood sampling reported by the authors. Similar data were reported by Doyle et al. [42]: 23 of 24 HCV-positive patients obtained SVR without significant variation of their HOMA-IR.

Regarding the impact of DAAs on the incidence of DM (Table 2), recent studies performed on large series of patients of the US Veterans Administration reported conflicting data; in the study of Butt and colleagues [43], 21279 diabetic patients treated with DAAs and 4764 nondiabetic patients treated with pegylated interferon (PEG-IFN)/ribavirin (RBV) were included, as well as a control group composed of untreated nondiabetic patients. The incidence of DM was recorded, although without providing the follow-up data, in the various subgroups and a significant reduction of DM onset was observed in patients treated with DAAs (9.89/1000 person-years, 95% confidence interval (CI) 8.7–11.1) compared to untreated patients (20.6, 95% CI 19.6–21.6, *p* < 0.001) and those treated with PEG-IFN/RBV (19.8, 95% CI 18.3–21.4), *p* < 0.001). The difference between the two types of treatment can be explained by the different efficacy rates of viral eradication obtained with IFN compared to IFN-free treatment. Moreover, these rates were not reported by the study, which merely describes an overall SVR rate of 78.82%. In multivariate analysis, the reduction in the risk of developing DM in patients treated with DAAs was quantified as 47% (relative risk (RR) 0.53, 95% CI 0.46–0.63, *p* < 0.0001) and the unfavorable predictive factors were ethnic origin (black or Hispanic), male sex, high BMI, and advanced liver fibrosis. Surprisingly, the results of this study contrast with those of a study [44] performed on identical series (only veterans, predominantly male) composed of 41711 patients with SVR and 3549 nonresponders after therapy with DAAs. After a mean follow-up of about 2 years, no statistically significant difference regarding the incidence of DM between the two groups was observed (21.04/1000 patients per year versus 23.11/1000 patients per year, hazard ratio (HR) = 0.98, 95% CI = 0.81–1.19, *p* = 0.86). Of interest, an American study [45] conducted in the IFN era on similar patient series showed that viral clearance had a significant impact on the incidence of diabetic disease. Like all retrospective studies performed on large databases, these studies have limitations: identification of co-factors influencing the onset of DM was not optimal; in particular, variations of lifestyle (diet, physical exercise) or habits (smoking) were not recorded. A hypothesis to explain the discrepancy could be the difference in the duration of the average follow-up, which was much longer in the study by Mahale et al. [45]. It is therefore likely that, with comparable follow-up, the results might not have been all that different, given that nonresponders generally show a strong tendency toward DM onset with the worsening of liver fibrosis.

Other studies on less selected and therefore more representative series showed a significant reduction of the onset of de novo DM in HCV-positive patients with SVR. In the study of Li and colleagues [46], 5127 nondiabetic patients treated with DAA were followed up: the incidence of DM after an average follow-up of 3.7 years was found to be 6.2% in sustained responders compared to 21.7% in nonresponders (HR = 0.79; 95% CI: 0.65–0.96). A study [47] conducted on liver transplant patients with HCV-positive cirrhosis showed that SVR obtained with DAAs induced a significant reduction in post-transplantation DM (HR = 0.40, *p* = 0.048). However, this decrease was found to be particularly marked in patients with SVR obtained before transplantation (9.7% versus 23%, HR = 0.30, *p* = 0.046), which adds another element in favor of the proposal to administer antiviral therapy before transplantation rather than after.

In conclusion, studies that evaluated the effect of DAA-induced SVR on the incidence of IR and DM in sustained responders seem to confirm what was already observed in the IFN era. However, it should be noted that comparing the effects of IFN treatment with those of DAAs is not relevant since they represent two completely different mechanisms of viral eradication. Last but not least, although the majority of the above-mentioned studies indicate a beneficial effect of SVR in reducing the risk of onset of IR/DM, appropriate follow-up is still lacking. Therefore, definite conclusions cannot be drawn on the lasting efficacy of antiviral therapy with DAAs with regard to the incidence of glycometabolic abnormalities.

Data coming from further follow-up studies will be crucial to determine beyond any reasonable doubt whether viral eradication significantly reduces the onset of IR and DM, resulting in an important benefit to global public health.

### 3.2. Does HCV Clearance with DAAs Lead to Significant Improvement of Glycometabolic Control in Patients with DM? If so, is This Control Maintained over the Long Term?

A recent review with a meta-analysis [28] demonstrated and quantified the improvement of glycometabolic control in diabetic patients with CHC achieving SVR after treatment with DAAs. Eleven publications were considered and discussed by the review but only five met the inclusion requirements for the meta-analysis. Regarding the variation of HbA1c levels, an average decrease of –0.45% was found (95% CI: –0.60 to –0.30%, *p* < 0.001) with strong heterogeneity between the studies (χ^2^ = 20.4, *p* < 0.001). Glycemic values were evaluated by three studies [15,20,21]: A significant reduction of –22 mg/dL (95% CI: –41.61 to –2.44 mg/dL, *p* = 0.03) was observed, with wide heterogeneity among the studies (χ^2^ = 35.82, *p* < 0.001). According to the reported data, the number of published studies that agreed with the glycometabolic amelioration induced by SVR was higher than the number of studies that disagreed with this hypothesis.

Actually, several studies (Table 3) show significant improvement in glycemic levels and HbA1C at the end of DAA therapy, compared to a few (Table 4) that did not show any significant change in glycometabolic control.

Two studies [18,19], although demonstrating significant glycometabolic improvement at SVR, did not confirm this improvement over the long term.

The number of recruited patients is extremely variable, ranging from a minimum of 13 to a maximum of 2435 patients, and methods regarding outcome evaluation vary. The follow-up is not homogeneous: it ranges from a minimum of 12 weeks to a maximum of 60 months. A decrease of HbA1c of at least 0.5% when compared to baseline values is usually considered a significant improvement of the glycemic state [48], but this cutoff point was adopted by only three studies [17,20,21].

In the study by Ciancio et al. [15], baseline glycemic and HbA1c levels as well as the type of antidiabetic therapy were comparable between the sustained virologic responders (group 1) and non-SVR-/untreated patients (group 2). Diabetic patients with SVR showed a significant decrease in glucose (152.4 ± 56.4 mg/dL versus 134.3 ± 41.3 mg/dL, *p* = 0.002) and HbA1c (6.9% versus 6.4%, *p* < 0.001) values at 12 weeks from the end of therapy; no significant glucose (145.3 ± 30.2 versus 140.0 ± 47.9 mg dL, *p* = 0.710) and HbA1c (7.0% versus, 7.2%, *p* = 0.780) reduction was observed among group 2 patients. Abdel Alem and colleagues [20] found a significant improvement of glycemic (103 mg/dL versus 113 mg/dL, *p* = 0.005) and HbA1c (6.4% versus 6.9%, p < 0.001) levels at 24 weeks from the end of therapy. Subgroup analysis showed that the decrease in glucose values occurred mainly in patients treated with oral hypoglycemic agents (100 mg/dL versus 116.5 mg/dL, *p* = 0.008) while decreased HbA1c was present in both insulin-treated and oral therapy patients (6.2% versus 6.7%, *p* = 0.007 and 6.5% versus 7.0%, *p* = 0.001, respectively). Dawood et al. [21] stratified diabetic patients with SVR (378 patients, 94.5%) into two groups based on the presence (292 patients, 77.2%) or absence (86 patients, 22.8%) of glycometabolic improvement to assess baseline predictive factors. Improvement was defined as a reduction in fasting blood glucose of at least 20 mg/dL or an HbA1c level of at least 0.5%. Genetic predisposition, advanced stage of liver fibrosis, and duration of diabetic disease were negative predictive factors for glycometabolic amelioration. After three months of DAA therapy, average glycemic and HbA1c levels decreased from 184.5 ± 27.9 to 136.5 ± 22.5 mg/dL and from 8.1 ± 0.4% to 7.3 ± 0.3%, respectively, in the group with significant glycometabolic improvement. In the study by Gilad et al. [17], a decrement of 0.6% (95% CI: 0.2–0.9, *p* < 0.01) in HbA1c was found 1.5 years after the end of therapy with DAAs (8.4% ± 1.9% versus 7.8% ± 1.7%). By defining glycometabolic improvement as a reduction of HbA1c (with or without reduction of antidiabetic therapy) or antidiabetic therapy (even without a significant variation of HbA1c), the authors showed an amelioration rate of 34%. The predictors of glycometabolic improvement were the presence of at least one antidiabetic therapy, the type of antidiabetic therapy (insulin), and higher creatinine levels. Hum and colleagues [14] retrospectively evaluated 2435 diabetic patients with HCV treated with DAAs; viral eradication was obtained in about 90% of cases and 255 patients were nonresponders. A significant decrease in HbA1c levels was observed only in patients with SVR and high pretherapy levels of HbA1c (>7.2%) compared to nonresponders (0.98 ± 1.4% versus 0.65 ± 1.5%, *p* = 0.02). In sustained responders with pretherapy HbA1c levels < 7.2% and in cirrhotic patients, no significant reduction was observed. These conclusions cannot be easily reproducible as the patients were American veterans, predominantly obese (average BMI > 30) males (98%) with genotype 1 (>99%). Limiting factors of the study were the inability to verify any other therapies besides the antidiabetic one and the impossibility of identifying any changes in lifestyle during follow-up, with all factors potentially influencing glycometabolic control. Pavone et al. [27] measured glycemic and HbA1c levels before and during treatment with DAAs in 29 diabetic patients, demonstrating an average reduction in fasting plasma glucose of 52.86 mg/dL (*p* = 0.007) and HbA1c of −1.95% (*p* = 0.021). Fabrizio and colleagues [25] confirmed these results with a larger series (59 patients) and a more robust follow-up, but evaluating only the glycemic values. The same approach was adopted by Drazilova and colleagues [16] who studied glycemic trends in a series of Slovak patients treated with DAAs: only diabetics or patients with impaired fasting glucose (IFG) showed a significant decrease in fasting plasma glucose (FPG) after achieving SVR. Ikeda et al. [23] measured HbA1c levels in 36 Japanese patients treated with DAAs before and after therapy; of these, 13 were diabetic. Globally, HbA1c levels decreased from 5.85% to 5.65% (*p* < 0.01), but the study did not report glycometabolic data concerning diabetic patients only. Morales et al. [22] retrospectively analyzed 60 patients with HCV treated and cured with sofosbuvir-based combination therapy (IFN was also used in 35%). Out of 60 patients, 23 (38.3%) were diabetic; data on HbA1c trends were available in 39 patients. A significant decrease of this parameter was observed 3–6 months after therapy (6.13% versus 6.66%, *p* < 0.005) without a significant average weight loss. Data regarding HbA1c changes were not reported according to the presence/absence of DM, but the authors showed a larger drop in HbA1c among diabetic compared to nondiabetic patients, although this was not statistically significant. Weidner and colleagues [18] evaluated over time glycemic and HbA1c values of 281 HCV-RNA-positive patients treated with DAAs, 28 of which had DM. Significant reductions in glycemia (from 168 ± 8.7 mg/dL to 146 ± 9.2 mg/dL, *p* = 0.04) and HbA1c (*p* = 0.0367) were observed in diabetic patients with SVR 6 months after the end of therapy, but this effect was observed only in noncirrhotic patients. Finally, Pashun et al. [26] published a case report regarding an obese woman with chronic hepatitis C and DM poorly controlled with insulin therapy, showing a significant decrease in HbA1c levels and a reduction in insulin dosage after DAA-induced SVR, maintained up to 15 months after the end of therapy, despite the observed weight increase.

The effect of antiviral therapy with DAAs on diabetic patients has also been studied in the liver transplant setting [24,29,30]. Beig and colleagues [24] observed a significant decrease of HbA1c levels (from 35.5 ± 4.3 mmol/mol to 33.3 ± 3.6 mmol/mol, *p* = 0.03) at 44 weeks from the end of treatment in 91 liver transplant recipients with recurrent HCV infection treated with DAAs. Unfortunately, it is not possible to establish from the reported data the HbA1c reduction in the 26 diabetic patients treated and cured by DAAs, even though the authors report a 40% decrement of DM treatment. These conclusions were not confirmed by Saab and colleagues [29] in 30 transplanted diabetic patients treated with DAAs; no statistically significant differences in glycemic values were observed before antiviral therapy and during follow-up. HbA1c levels were not evaluated in this study. In the study by Teegen et al. [30], 32 HCV-positive diabetic patients on insulin therapy who underwent orthotopic liver transplantation (OLT) achieved SVR after DAA therapy; HbA1c levels were monitored from baseline until the end of follow-up (30.6 months), with a nonstatistically significant decrease (from 7.1% to 6.4%, *p* = 0.098). However, two patients discontinued insulin therapy and a decrease in the cumulative insulin dosage (from 55.3 U/d to 38.2 U/d, *p* = 0.009) was recorded at the end of treatment in the absence of weight changes.

In addition to the studies by Saab et al. [29] and Teegen et al. [30], other publications [31,32,33] showed no significant improvement in glycometabolic control after viral eradication with DAAs. Of these, two recruited a mixed cohort of diabetic and nondiabetic patients [32,33]; in one case, glycemic and HbA1c levels referred to the entire cohort [33]. Stine et al. [31] did not show any significant changes in HbA1c levels 12 weeks after SVR in 26 diabetic patients treated with DAAs (from 7.3% to 7.1%, *p* = 0.268). Furthermore, 31% of patients had to increase their dosage of insulin therapy. Chaudhury and colleagues [32] found no significant differences between baseline and follow-up values of glycemia and HbA1c in sustained responders and nonresponders. Of the 42 patients with SVR, only 7 (3%) reduced their dosage of antidiabetic therapy. Huang et al. [33] evaluated HbA1c levels in a mixed cohort of diabetic (13, 21.7%), prediabetic (11, 18.3%), and normoglycemic (36, 60%) patients before starting therapy and 12 weeks after the end of treatment without finding significant variations (from 5.6 ± 0.6% to 5.5 ± 0.6%, *p* = 0.17). These three studies show important methodological limitations; the number of patients recruited was rather low (range 13–42); in one study [32], it is not possible to extrapolate the glycemic and HbA1c values according to the presence of DM or not; and, despite the reported absence of glycometabolic improvement, the reduction of antidiabetic therapy was observed only in diabetic patients with SVR.

Once it is established that most of the current evidence agrees that SVR induces a significant improvement in glycometabolic control in a substantial proportion of patients, the next question is whether this effect is maintained over the long term. Data addressing this question are scant, heterogeneous, and often contradictory. Most studies [15,16,20,21,22,23,25,27,31,41] report an average follow-up ranging from end of treatment to 24 weeks after discontinuation of therapy; publications [14,17,18,19,24,26,30,32] with longer follow-up (>24 weeks from the end of the therapy) provide discrepant data. In the case report by Pashun et al. [26], HbA1c levels continued to improve until the last follow-up check (15 months after the end of treatment) despite weight gain and progressive tapering of insulin therapy. In the study of Hum and colleagues [14], glycometabolic improvement involved only noncirrhotic diabetic patients with pretherapy levels of HbA1c ≥ 7.2%, but this improvement was maintained up to 15 months after the end of treatment. Gilad et al. [17] followed their patients with SVR up to 1.5 years post-therapy and observed maintenance of the reduction in HbA1c levels, with no differences between cirrhotic and noncirrhotic patients; in particular, in the secondary analysis regarding patients who improved from a diabetological point of view (≥0.5% reduction in HbA1c levels with or without modification of antidiabetic therapy or reduction/suspension of antidiabetic therapy in the absence of significant changes in HbA1c), 71% showed sustained improvement. Similar data were reported by Beig and colleagues [24] in a population of liver transplant recipients: maintenance of glycometabolic improvement in 38 diabetic patients was observed up to 44 months after obtaining SVR, with sustained cessation of antidiabetic therapy in about 40% of patients. According to Weidner et al. [18], the significant reduction in HbA1c levels was maintained until the 24th follow-up month only in noncirrhotic diabetic patients, whereas in cirrhotic diabetic patients, this improvement disappeared from the 12th month of follow-up. It should be noted that of the 28 diabetic patients considered, only 22 had baseline data and of these, only 9 were followed up until 24 months after the end of therapy. Of the 73 diabetic patients treated with DAAs included in the study of Li and colleagues [19], 66 (90.4%) achieved SVR; in the 7 nonresponding diabetic patients and the 73 untreated HCV-positive diabetic patients who acted as a control group, HbA1c levels remained unchanged over the long term, while in the 66 patients with SVR, a triphasic course was observed: a significant reduction up to 6 months from the end of therapy, a rebound in the following months, and a stabilization of the pretherapy levels from the 30th to 60th follow-up month. The retrospective/prospective study design was rather complex, with some limitations: in the general analysis, patients treated with IFN were also included and there were no data regarding possible confounding factors such as changes in lifestyle and adherence to antidiabetic therapy. Chaudhury et al. [32] prospectively followed 42 HCV-RNA-positive diabetic patients treated and cured with DAAs (except one), with a mean follow-up of 28 months after the end of therapy, and compared the mean HbA1c values in sustained responders before treatment and at the end of follow-up, without observing significant modifications of this parameter. Similar results were obtained by Teegen and colleagues [30] in a population of transplanted patients with a mean follow-up of 30.6 months.

In conclusion, an improvement of glycometabolic control at the end of therapy or in the immediate post-therapy months in diabetic patients, or only in some subgroups, as reported by Hum and colleagues [14] and by Weidner et al. [18], is reported by the vast majority of studies published so far. Whether this beneficial effect is maintained over the long term is still a matter of debate. For this reason, large prospective cohort studies with appropriate analysis of possible confounding factors (smoking status, steatosis staging, BMI variations, physical exercise, adherence to antidiabetic therapy) are urgently needed to establish the persistence of such an amelioration. Meanwhile, we can hypothesize that subgroups of patients will benefit from viral eradication in the long term also on the metabolic side, but on the other hand, it is reasonable to think that a significant proportion of diabetic patients with SVR, probably patients with advanced cirrhosis and/or a long duration of diabetic disease, will not experience lasting benefits from viral clearance and the effectiveness of glycometabolic control will have to be based on lifestyle and adherence to antidiabetic therapy. This observation suggests a point of no return where the impact of viral eradication on the outcome of diabetic disease is minimal or irrelevant; therefore, it is extremely important to quickly identify HCV-positive diabetic patients in order to start antiviral therapy without delay, so as to stop fibrotic progression, on the one hand, and to reduce the risk of the occurrence of extrahepatic complications related to DM on the other.

### 3.3. Does SVR-Related Glycometabolic Improvement Determine De-Escalation/Withdrawal of Antidiabetic Therapy?

Few studies have analyzed this aspect (Table 5) and often do not specify the type of modified treatment or criteria leading to the reduction/withdrawal.

In the study by Ciancio et al. [15], antidiabetic therapy was reduced, changed, or even suspended only in patients with SVR (20.7%, *p* = 0.03). In detail, for 8 of the 19 patients (42.1%) taking oral hypoglycemic therapy and 13 of the 46 (28.2%) on insulin therapy, administration of the drug was decreased or suspended. It should be noted that these therapeutic variations occurred in the presence of an average weight gain of 2.85 ± 4.33 kg among sustained responders. Of the 101 patients with SVR, 9 (8.9%) had to have their therapeutic dosage increased compared to 1 nonresponder/untreated patient (4.8%, *p* = 0.34). According to Dawood et al. [21], dosage reduction was observed only in cured patients with concomitant improvement in glycometabolic control (292 patients). In this group, 78 patients (26.7%) had their antidiabetic therapy reduced (61 patients on insulin therapy, 17 treated with gliclazide). Similar data were obtained by Pavone and colleagues [27]: 23% of patients with SVR and glycemic amelioration had their antidiabetic treatment de-escalated. In detail, of the 29 diabetic patients, 9 were treated with metformin, 1 with repaglinide, and 15 with insulin, while the remaining 4 were on diet therapy only. At the end of therapy, 2 of the 9 patients treated with metformin and 4 of the 15 on insulin therapy had their dosage reduced. Of the 13 diabetic patients included in the study by Ikeda et al. [23], 6 were treated with insulin, 6 with oral hypoglycemic agents, and 1 with diet alone. Of the 12 patients on therapy, 3 (25%) had their pharmacological dosage reduced (1 on insulin therapy and 2 treated with oral hypoglycemic agents). Hum and colleagues [14] showed that the percentage of insulin-treated diabetic patients significantly decreased among those with SVR (from 41.3% to 38%) compared to the control group (from 49.8% to 51%, *p* = 0.04). This reduction was not accompanied by an increase in the proportion of patients treated with metformin, which, on the contrary, was also reduced by 2.2%.

Regarding patients with OLT [24], of the 26 diabetic patients on antidiabetic therapy before antiviral treatment, 24 received insulin and 2 insulin + oral hypoglycemic agents. At the end of follow-up, only 16 received antidiabetic therapy, with suspension of this therapy in the remaining 10 (38%). Fabrizio and colleagues [25] did not show significant therapeutic variations despite the reduction in glycemic values: of the 59 diabetic patients considered, 49 were on antidiabetic therapy (16 on insulin, 31 on oral hypoglycemic agents, 2 on both drugs), but only 1 patient on insulin therapy had the dosage reduced, while 2 of 4 patients with worsening glycemic control had to have their dosage increased.

In the study by Morales et al. [22], 2 of 16 diabetic patients (12%) receiving antihyperglycemic medications had their insulin therapy de-escalated or suspended, but it is not known how many were on insulin therapy and how many were on oral hypoglycemic agents. Of the 30 diabetic patients on antidiabetic therapy recruited in the study of Drazilova et al. [16], 13 were given oral hypoglycemic agents and 17 insulin. No reduction in oral therapy was observed, whereas 3 of 17 (17.6%) insulin-treated patients had to have their dosage reduced for documented episodes of hypoglycemia. Chaudhury and colleagues [32] and Stine and colleagues [31] did not show any significant therapeutic modifications concerning antidiabetic treatment. On the contrary, in the study by Teegen et al. [30] conducted on OLT patients, although no significant decrease in HbA1c levels was found, 2 of 32 patients had their insulin therapy suspended and a significant reduction in cumulative insulin dose was observed (55.3 versus 38.2 U/d, *p* = 0.009).

In conclusion, most of the studies reporting improvements in glycometabolic control showed reduction/suspension of antidiabetic therapy in a significant minority of patients; for this reason, it has been suggested that diabetic patients with DAA-induced SVR should be carefully monitored regarding their glycometabolic profile in order to avoid hypoglycemic episodes [15]. However, the heterogeneity and incompleteness of the data do not allow us to assess the characteristics of the patients who obtained this reduction/suspension or to identify a particular class of drugs particularly sensitive to this variation. Furthermore, the brief follow-up reported by the studies does not permit us to determine whether this modification was maintained over the long term. Future work will need to precisely evaluate antidiabetic therapy modification in more depth as it represents a clinically meaningful endpoint.

### 3.4. Do DAA-Induced Glycemic Changes Determine a Significant Clinical Impact on the Outcome of DM and its Complications?

During the IFN era, some studies [1,49,50] showed a significant decrease in overall liver- and non-liver-related mortality in patients achieving SVR. This finding was recently confirmed by a French prospective study [51] on 9895 patients with chronic hepatitis C, with 7344 treated with DAAs and 2551 untreated. At the end of follow-up, significant reductions in mortality (from any cause) and in the incidence of hepatocellular carcinoma (HCC) in treated patients compared to untreated were found. The decreased mortality from extrahepatic causes was probably due to the reduction of HCV-related extrahepatic manifestations observed among patients with SVR [13].

Since IR and DM are now considered extrahepatic manifestations of HCV, a reduction in complications of DM can be expected in diabetic patients with viral clearance. A Taiwanese study [52] on diabetic patients with CHC treated with IFN showed a significant reduction in DM-related complications, such as nephropathy, acute coronary artery disease, and stroke. However, several methodological limitations of the study (exclusion of patients with significant co-morbidities, no data on SVR, lack of details regarding patient characteristics and glycometabolic control) may have influenced the results. Recently, an American study [53] on 723 diabetic patients and CHC treated with IFN or DAAs showed a significant decrease in the incidence of acute coronary syndrome (HR = 0.36, *p* < 0.001), advanced nephropathy (HR = 0.46, *p* < 0.001), stroke (HR = 0.34, *p* < 0.001), and retinopathy (HR = 0.24, *p* < 0.001) in SVR patients compared to noncured or untreated patients. However, some methodological weaknesses (lack of data on cardiovascular risk factors including smoking, cholesterol, hypertension, adherence to diet, exercise, antiplatelet drugs) and the short duration of follow-up (2.4–2.7 years) suggest caution regarding the reproducibility of the study. Moreover, in a previous study [19], the authors showed that glycometabolic amelioration was not maintained over the long term. How is it possible to achieve a significant decrease of DM-related complications without a parallel persistent glycometabolic improvement?

A possible explanation could derive from the observation that viral clearance with DAAs determines a significant improvement of extrahepatic diseases such as cardiovasculopathies and nephropathies, regardless of the presence of DM. A recent American study [54] showed a significant improvement in extrahepatic manifestations accompanying C virus infection in patients with CHC cured after therapy with DAAs. The percentage of diabetics in this population was less than 10%. In an Italian study [55], a significant amelioration of carotid atherosclerosis, intima-media thickness, and carotid thickening was found in 182 patients treated with DAAs; 20% were DM carriers. Improvement was not influenced by cardiovascular risk factors such as DM, hypertension, hypercholesterolemia, and smoking. These data suggest that the improvement of extrahepatic manifestations of HCV infection (in particular, cardiovasculopathies and nephropathies) coinciding with the most frequent complications of diabetic disease may improve after viral clearance due not only to glycometabolic control amelioration, but also to other mechanisms yet to be established. In the case of nephropathy, it is likely that HCV eradication may lead to an improvement in vasculitis–cryoglobulinemic damage [56,57]; the beneficial effect of viral eradication on cardiovascular diseases may be derived from the disappearance of the systemic inflammatory status [58], the decrease in TNF-α and IL-6 levels [59,60], and the increase in adiponectin levels [61].

In conclusion, it is still unclear whether the reduction in complications of diabetic disease observed after viral eradication is mainly due to glycometabolic improvement or to the direct beneficial effect of viral clearance on extrahepatic sites. Further studies with appropriate follow-up are needed to assess which is the key mechanism and whether this beneficial effect is maintained over the long term. However, the data show that no improvement in DM is observed in patients with advanced fibrosis or with long-term DM history, suggesting that there is a point of no return for the modifications in glucose metabolism and, conversely, DM clinical outcome. For this reason, early identification of HCV-positive diabetic patients is strongly recommended in order to start DAA therapy as early as possible.

### 3.5. What is the Clinical Impact of DAA-Induced SVR on The Incidence of HCC in Diabetic Patients?

Previous studies [62,63,64] conducted on IFN-treated patients showed that viral eradication resulted in a significant reduction in liver-related events (hepatic decompensation, variceal bleeding, HCC) in cirrhotic patients. Similarly, two studies [65,66] on noncirrhotic HCV-positive patients with IFN-induced SVR demonstrated a decrease in the incidence of HCC. However, much evidence [63,64,65,66,67,68,69,70,71,72] suggests that DM is an independent predictor of HCC, despite viral eradication obtained by IFN-based therapies.

Data from follow-up studies on patients with SVR after therapy with DAAs [51,73,74,75,76,77,78,79,80,81] are mainly focused on the incidence of HCC in cured patients and are necessarily limited by the short follow-up. In most of these studies [51,73,74,75,76,77,78,79], DM was not found to be an independent predictor of HCC in patients with SVR; only two studies [80,81] confirmed the negative effect of diabetic disease on the development of HCC after viral clearance. Unfortunately, the lack of data on the glycometabolic control or the type of antidiabetic treatment limits the possibility of drawing definite conclusions.

Arase and colleagues [71] showed that maintaining HbA1c levels <7.0% during follow-up significantly reduced the onset of HCC in diabetic patients with IFN-induced SVR. Similarly, some studies [82,83,84] suggested a protective action of metformin therapy against HCC compared to other antidiabetic treatments; an American study [85] showed a significant increase in the survival of DM patients with cirrhosis of any etiology who did not have metformin therapy suspended or changed with other types of antidiabetic therapy during follow-up.

In conclusion, the magnitude of risk reduction in HCC incidence found among diabetic patients with SVR is probably comparable to that observed in nondiabetics; however, the discrepancy between the data of the IFN period compared with the DAA period is puzzling and deserves further consideration. It is likely that the difference in follow-up, which is generally longer in the IFN-based studies, may have influenced the final results. However, the metabolic influence of DM on the residual HCC risk in DAA-cured patients should be reconsidered thanks to recent studies [86,87] showing evidence of HCV-induced epigenetic changes associated with HCC risk. According to these studies, the viral epigenetic signature in the host genome is often irreversible, in particular in those patients with advanced liver fibrosis.

## 4. Conclusions

According to the vast majority of studies considered, the incidence of IR and DM appears to decrease in CHC patients achieving SVR after DAA treatment, but the extent of this reduction is questionable. In sustained responders with established DM, glycometabolic improvement is often found with concomitant reduction of antidiabetic therapy in a significant subset of patients. However, whether these beneficial effects are long lasting or just transient is still an unresolved issue; conversely, the long-term clinical implications in terms of diabetic and hepatic outcomes of these biochemical and pharmacological variations remain largely unknown. Therefore, a careful longitudinal monitoring of these patients is needed before claiming any consolidated efficacy. Obviously, in order to arrive at meaningful conclusions, it is necessary to await the results of long-term follow-up prospective studies with an adequate number of patients. These results should include not only the biochemical but also the clinical aspect; in other words, the main endpoint to be tested must focus on the question of whether any improvement in glycemic control is going to have a major impact on the outcome of liver and diabetic disease.

## Figures and Tables

**Figure 1 jcm-09-00563-f001:**
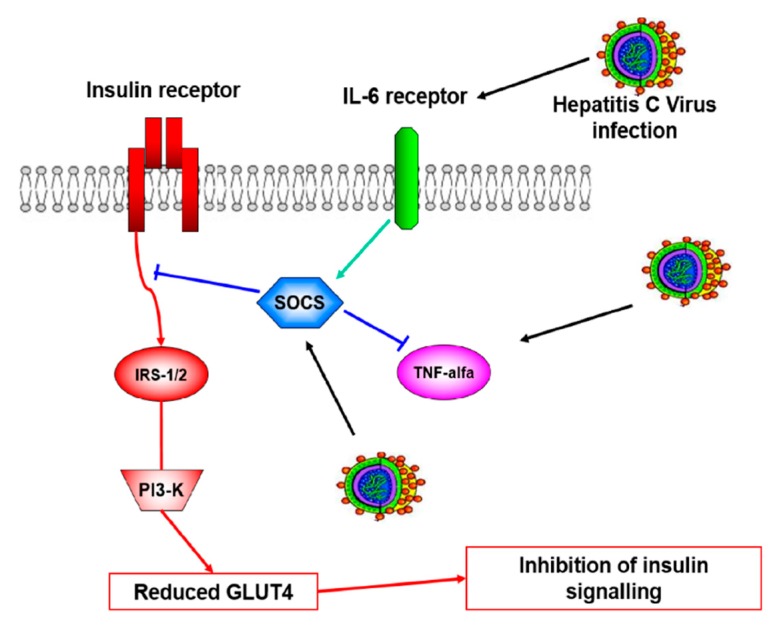
Theoretical explanation of the pathogenesis of hepatitis C virus-induced insulin resistance. The virus directly or indirectly, through the induction of interleukin 6 (IL-6) and/or tumor necrosis factor-alpha (TNF-α), determines the protoeosomal degradation of the insulin receptor substrate (IRS 1 and 2) by inducing suppression of cytokine signalling (SOCS); the consequent downregulation of phosphoinositide 3-kinase (PI3-K) causes reduced GLUT4 (glucose carrier) transmembrane expression, leading to the final inhibition of insulin signalling.

**Table 1 jcm-09-00563-t001:** Impact of DAA treatment on insulin resistance in nondiabetic HCV patients.

Author, Year (ref.)	Type of Study	Design	No. of Patients with SVR	No. of Control Patients	HOMA-IR Change (*p*)	Follow-up
Salomone et al. 2018 [36]	Observational	Prospective	32	N.A.	−1.4 (<0.001)	12 weeks
Elhelbawy et al. 2019 [37]	Observational	Prospective	465 (24% with DM)	46	−1.09 (<0.001)	12 weeks
Adinolfi et al. 2018 [38]	Case-control	Prospective	68	65	−2.66 (<0.001)	12 weeks
Gualerzi et al. 2018 [39]	Observational	Prospective	82	N.A.	−0.62 (<0.001)	12/24 weeks
Andrade et al. 2018 [40]	Observational	Prospective	75	N.A.	−0.76 (0.02)	12 weeks
Meissner et al. 2015 [41]	Phase 2 clinical trial	Prospective	33	17	Not significant	36 weeks
Doyle et al. 2019 [42]	Observational	Prospective	23	N.A.	−0.3 (0.32)	24 weeks

DAA, direct-acting antiviral agent; HCV, hepatitis C virus; SVR, sustained virologic response; N.A., not available; HOMA-IR, homeostatic assessment of insulin resistance; DM, diabetes mellitus.

**Table 2 jcm-09-00563-t002:** Incidence of DM in HCV patients treated with DAAs.

Author, Year (Reference)	Type of Study	Design	No. of Patients with SVR	No. of Control Patients	DM Incidence in SVR Patients	DM Incidence in Control Patients	*p*-Value	Follow-up
Butt et al. 2019 [43]	Observational	Retrospective	21279	4764	9.9/1000 persons/year	20.9/1000 persons/year	<0.001	6 years
El-Serag et al. 2019 [44]	Observational	Retrospective	41711	3549	21.0/1000 persons/year	23.1/1000 persons/year	0.86	2 years
Li et al. 2018 [46]	Observational	Retrospective	3748	1397	6.2%	21.7%	0.0003	3.7 years
Roccaro et al. 2019 [47]	Observational	Retrospective	31	225	9.7%	23,0%	0.02	41,2 months

DM, diabetes mellitus; HCV, hepatitis C virus; DAAs, direct acting antiviral agents; SVR, sustained virologic response.

**Table 3 jcm-09-00563-t003:** Studies reporting significant glycometabolic improvement in DAA-treated HCV-positive patients with SVR.

Author, Year (Reference)	Type of Study	Design	No. of Diabetic pts	Mean Glycemic Change (*p*)	Mean HbA1c Level Change (*p*)	Follow-up
Ciancio et al. 2018 [15]	Observational	Prospective	101	−18.0 md/dL (0.002)	−0.5% (<0.001)	12 weeks
Abdel Alem et al. 2017 [20]	Observational	Retrospective	65	−11.5 mg/dL (0.005)	–0.5% (<0.001)	24 weeks
Dawood et al. 2017 [21]	Clinical trial	Open label	378	−23.4 mg/dL (N.A.)	−0.45% (N.A.)	12 weeks
Bejg et al. 2018 [24]	Observational	Retrospective	38	–19.8 mg/dL (0.01)	−0.2% (0.03)	24–44 weeks
Pavone et al. 2016 [27]	Observational	Retrospective	27	−52.7 mg/dL (0.0007)	−2.0% (0.02)	8 weeks
Weidner et al. 2018 [18]	Observational	Retrospective	28	−22 mg/dL (0.04)	−0.29% (0.04)	0–44 weeks
Gilad et al. 2019 [17]	Observational	Retrospective	122	Not determined	−0.6% (0.001)	1.5 years
Hum et al. 2017 [14]	Observational	Retrospective	2435	Not determined	−0.37% (0.03)*	48 weeks
Morales et al. 2016 [22]	Observational	Retrospective	23	Not determined	−0.53% (<0.005)	24 weeks
Ikeda et al. 2017 [23]	Observational	Prospective	13	Not determined	−0.2% (<0.01)	12 weeks
Pashum et al. 2016 [26]	Case report	–	1	Not determined	−5.8% (N.A.)	24 weeks
Li et al. 2019 [19]	Observational	Retrospective/Prospective	192	Not determined	−2.3 (<0.001)	24 weeks
Fabrizio et al. 2017 [25]	Observational	Retrospective	59	–20 mg/dl (<0.001)	Not determined	24 weeks
Drazilova et al. 2018 [16]	Observational	Retrospective	88	−21 mg/dl (<0.0001)	Not determined	12 weeks

DAA, direct acting antiviral agent; HCV, hepatitis C virus; SVR, sustained virologic response; HbA1c, hemoglobin A1c; N.A., not available; * −0.13%, *p* = 0.01 when adjusted by multiple regression analysis.

**Table 4 jcm-09-00563-t004:** Studies reporting no significant glycometabolic improvement in DAA-treated HCV-positive patients with SVR.

Author, Year (Reference)	Type of Study	Design	No. of Diabetic pts	Mean Glycemic Change (*p*)	Mean HbA1c Level Change (*p*)	Follow-up
Stine et al. 2017 [31]	Observational	Retrospective	26	Not determined	−0.25% (0.27)	12 weeks
Saab et al. 2018 [29]	Observational	Retrospective	30	N.A. (0.32)	Not determined	27 months
Teegen et al. 2019 [30]	Observational	Retrospective	32	Not determined	−0.7% (0.1)	31 months
Chaudhury et al. 2018 [32]	Observational	Prospective	41	−6 mg/dl (0.21)	−0.1% (0.29)	28 months
Huang et al. 2017 [33]	Clinical trial	Prospective	13	Not determined	–0.1% (0.17)	3−15 months

DAA, direct acting antiviral agent; SVR, sustained virologic response; HbA1c, hemoglobin A1c; N.A., not available.

**Table 5 jcm-09-00563-t005:** Studies reporting de-escalation/withdrawal of antidiabetic therapy in HCV-positive diabetic patients with DAA-induced SVR.

Author, Year (Reference)	Type of Study	Design	De-Escalation/Withdrawal of Oral Hypoglycemic Agents, *n*/total (%)	De-Escalation/Withdrawal of Insulin Therapy, *n*/total (%)	Follow-up (EOT)
Hum et al. 2017 [14]	Observational	Retrospective	85/1630 (5.2%)	72/900 (8.0%)	12 months
Ciancio et al. 2018 [15]	Observational	Prospective	8/19 (41.2%)	13/46 (28.2%)	12 weeks
Dawood et al. 2017 [21]	Clinical trial	Open label	17/378 (4.4%)	61/378 (16.1%)	12 weeks
Morales et al. 2016 [22]	Observational	Retrospective	2/16 (12.5%)	N.A.	24 weeks
Ikeda et al. 2017 [23]	Observational	Prospective	2/6 (33.3%)	1/6 (16.6%)	12 weeks
Bejg et al. 2018 [24]	Observational	Retrospective	2/26 (7.6%)	10/26 (38.4%)	24–44 weeks
Fabrizio et al. 2017 [25]	Observational	Retrospective	0/31 (0.0%)	1/16 (6.2%)	12–24 weeks
Pavone et al. 2016 [27]	Observational	Retrospective	2/10 (20.0%)	4/15 (26.6%)	8 weeks
Drazilova et al. 2018 [16]	Observational	Retrospective	0/13 (0.0%)	3/17 (17.6%)	12 weeks
Teegen et al. 2019 [30]	Observational	Retrospective	N.A.	2/32 (6.25%)	30.6 months

HCV, hepatitis C virus; DAA, direct-acting antiviral agent; SVR, sustained virologic response; EOT, end of treatment; N.A., not available.

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
