# Peer review of "The Effect of Viral Clearance Achieved by Direct-Acting Antiviral Agents on Hepatitis C Virus Positive Patients with Type 2 Diabetes Mellitus: A Word of Caution after the Initial Enthusiasm"

_jcm, 2020, doi:10.3390/jcm9020563_

Round 1
Reviewer 1 Report
Comments to the editor
The review manuscript submitted by Davide Giuseppe Ribaldone and co-workers aims to summarize the biochemical and clinical consequences of hepatitis C virus clearance on diabetes mellitus type 2 (DM) and insulin resistance (IR), which is one of the extrahepatic manifestations on HCV infection in patients. Therefore, the authors performed a thorough extensive literature search focusing on published potential impacts of HCV cure (SVR) with direct-acting antivirals on DM. The review was structured to respond to 5 especially pertinent questions:
1) Does SVR achieved by DAAs significantly prevent the onset of IR and DM?
2) Does HCV clearance with DAAs lead to significant improvement of glycometabolic control in patients with DM? And if so, is this control maintained over the long term?
3) Does SVR-related glycometabolic improvement induce de-escalation/withdrawal of the anti-diabetic therapy?
4) Do DAAs-induced glycaemic changes determine a significant clinical impact on the outcome of DM and its complications?
5) What is the clinical impact of DAAs-induced SVR on the incidence of HCC in diabetic patients?
The review manuscript is generally very well written, comprehensive but concise, up-to date and well structured. The topic of the review is very relevant to the field since more and more evidence is beginning to emerge that HCV cure cannot fully eradicate the risk of complications in patients with advanced liver disease. This review helps to shed light into the important metabolic consequences of HCV infections and will be useful to design future studies to study DM and IR in HCV-cured patients. However, the biochemical aspect of DAA-cure on HCV-associated liver disease remains a bit weak and could be strengthened in the discussion section. Especially, since recent studies from teams in Israel and France showed evidence of an epigenetic viral footprint in the host genome that contributes to the observed residual HCC risk in DAA-cured patients (PMIDs 30836093, 31216276). It would be interesting to discuss this in context of extrahepatic manifestations, also since here the liver is a central regulator of systemic insulin resistance.
Author Response
Your suggestion has been met. The concept of an “epigenetic signature” (often irreversible) induced by HCV has been introduced in the discussion of the Question 5 (line 483-487). Pertinent references (86,87) have been added.

Reviewer 2 Report
While this study covers an important aspect of HCV-related comorbidities and has significant clinical importance, there are a number of points that need serious consideration.
1) The language style is recommended to be improved. Some terms are very inappropriate and confusing (e.g. chronic C virus, line 266, 447, percentage of improved patients 244-245, etc.).
2) the study presents the data in an overloaded, too extensive way that makes it hard to digest, commonly focusing on repeating the results of the multiple studies, without providing any discussion. Authors should synthesize their findings and provide exhaustive discussion.
3) Authors should provide a paragraph on limitations of their study. For example, Authors often compare the effects of IFN treatment with those of DAA: this is not eligible since they represent two completely different mechanisms of viral eradication. Furthermore, even within DAA-based schemes there is a wide range of regimens used, which also may have impact on the analyzed effect.
4) there are some serious mistakes in the text, e.g. lines 36-37: in IFN-based therapies, the SVR was routinely assessed 24 weeks and not 12 weeks post-therapy.
5) Table 2, no P values are provided.
6) some abbreviations are not explained, e.g. OLT (line 394), FPG (line 262), while some are used erroneously, e.g. CHC means chronic hepatitis C and not chronic hepatitis C virus (line 27). Furthermore, HCV infection is not equal to CHC, since viral infection may be present in the absence of clinical hallmarks of hepatitis.
7) the Authors could provide more information on patomechanism of IR and DM induction in HCV infection and on how viral eradication by means of DAA-treatment could lead to resolution of these comorbidities. This could be accompanied by an appropriate Figure, as there is none.
8) finally, the study is largely inconclusive, since after exhaustive description of studies, no definite conclusions could be drawn.
Author Response
1) The language style is recommended to be improved. Some terms are very inappropriate and confusing (e.g. chronic C virus, line 266, 447, percentage of improved patients 244-245, etc.).
The text has been revised by a native English. Inappropriate and/or confusing terms were corrected and changes highlighted.
2) the study presents the data in an overloaded, too extensive way that makes it hard to digest, commonly focusing on repeating the results of the multiple studies, without providing any discussion. Authors should synthesize their findings and provide exhaustive discussion.
Your suggestion has been met. Several redundant parts of the text have been deleted (previous line 200 up to line 203, previous line 204 up to line 207, previous line 212 up to line 221).
Discussion of results has been improved (line 400 up to 402, line 406-408, line 454 up to 458, line 483 up to 487)
3) Authors should provide a paragraph on limitations of their study. For example, Authors often compare the effects of IFN treatment with those of DAA: this is not eligible since they represent two completely different mechanisms of viral eradication. Furthermore, even within DAA-based schemes there is a wide range of regimens used, which also may have impact on the analyzed effect.
Your suggestion has been met. A sentence regarding inappropriate comparison between IFN effects and DAAs effects has been added (line 166-168)
4) there are some serious mistakes in the text, e.g. lines 36-37: in IFN-based therapies, the SVR was routinely assessed 24 weeks and not 12 weeks post-therapy.
The definition has been deleted (line 37).
5) Table 2, no P values are provided.
P values were added.
6) some abbreviations are not explained, e.g. OLT (line 394), FPG (line 262), while some are used erroneously, e.g. CHC means chronic hepatitis C and not chronic hepatitis C virus (line 27). Furthermore, HCV infection is not equal to CHC, since viral infection may be present in the absence of clinical hallmarks of hepatitis.
Explanations of OLT and FPG have been added (line 275-276 and line 248) . We acknowledge the difference between HCV and CHC; terms have been corrected (line 27 and 29).
7) the Authors could provide more information on patomechanism of IR and DM induction in HCV infection and on how viral eradication by means of DAA-treatment could lead to resolution of these comorbidities. This could be accompanied by an appropriate Figure, as there is none.
Your suggestion has been met. Figure 1 (line 33) with its legend (page 19) explaining the theoretical pathogenesis of HCV-induced IR has been added.
8) finally, the study is largely inconclusive, since after exhaustive description of studies, no definite conclusions could be drawn.
We completely agree with you. This is the real hearth of the question: no definite conclusions could be drawn but this is not our fault but it is due to the lack of solid prospective data. When present, they are contradictory or conflicting particularly when the long term effect of SVR on DM is considered. This finding led us to choose the last part of the paper’s title: “..a word of caution after the initial enthusiasm”; according to us, several investigators were misled by the intial effects of SVR on glycemic/HbA1c values not considering the possibility of a transient effect. We think that a review warning hepatologists on the necessity of solid prospective long-term data before announcing the long term efficacy of SVR on DM is not futile.

Reviewer 3 Report
Very interesting paper concerning an important clinical issue. May be published
Author Response
Thank you for appreciating our review.
Round 2
Reviewer 2 Report
The manuscript has improved, but still would really benefit from a thorough language editing.
Author Response
Dear Reviewer,
Thank you for considering our paper improved.
Please find enclosed a copy of the new version of our article, revised by MDPI English editing service.
Yours sincerely,
Giorgio Maria Saracco